# Assembly and Analysis of the Complete Mitochondrial Genome of *Capsella bursa-pastoris*

**DOI:** 10.3390/plants9040469

**Published:** 2020-04-08

**Authors:** Denis O. Omelchenko, Maxim S. Makarenko, Artem S. Kasianov, Mikhail I. Schelkunov, Maria D. Logacheva, Aleksey A. Penin

**Affiliations:** 1Institute for Information Transmission Problems of the Russian Academy of Sciences, 127051 Moscow, Russia; artem.kasianov@gmail.com (A.S.K.); shelkmike@gmail.com (M.I.S.); maria.log@gmail.com (M.D.L.); alekseypenin@gmail.com (A.A.P.); 2Skolkovo Institute of Science and Technology, 121205 Moscow, Russia

**Keywords:** *Capsella bursa-pastoris*, complete mitochondrial genome, SMRT PacBio, structural variants, RNA editing

## Abstract

Shepherd’s purse (*Capsella bursa-pastoris*) is a cosmopolitan annual weed and a promising model plant for studying allopolyploidization in the evolution of angiosperms. Though plant mitochondrial genomes are a valuable source of genetic information, they are hard to assemble. At present, only the complete mitogenome of *C. rubella* is available out of all species of the genus Capsella. In this work, we have assembled the complete mitogenome of *C. bursa-pastoris* using high-precision PacBio SMRT third-generation sequencing technology. It is 287,799 bp long and contains 32 protein-coding genes, 3 rRNAs, 25 tRNAs corresponding to 15 amino acids, and 8 open reading frames (ORFs) supported by RNAseq data. Though many repeat regions have been found, none of them is longer than 1 kbp, and the most frequent structural variant originated from these repeats is present in only 4% of the mitogenome copies. The mitochondrial DNA sequence of *C. bursa-pastoris* differs from *C. rubella*, but not from *C. orientalis*, by two long inversions, suggesting that *C. orientalis* could be its maternal progenitor species. In total, 377 C to U RNA editing sites have been detected. All genes except *cox1* and *atp8* contain RNA editing sites, and most of them lead to non-synonymous changes of amino acids. Most of the identified RNA editing sites are identical to corresponding RNA editing sites in *A. thaliana*.

## 1. Introduction

Shepherd’s purse (*Capsella bursa-pastoris*) is a small herbaceous plant of the mustard family (Brassicaceae) and one of the most common weeds growing in diverse habitats on almost every continent. Being a recent allotetraploid and a close relative of the well-known model plant *Arabidopsis thaliana,* it is a promising model plant for studying polyploidization and its role in the adaptation and evolution of the flowering plants [1,2].

Mitochondria are important organelles that provide energy conversion from the food fuel molecules into cell usable ATP energy storage molecules in eukaryotic cells. Mitochondrial genomes (mitogenomes) are a valuable source of genetic information for phylogenetic studies and the investigation of essential cellular processes. Plant mitogenomes are highly variable molecules in both size and structure, in contrast to chloroplast genomes that have highly conserved quadripartite structure among land plants [3]. Mitochondrial genomes vary greatly not only between species but sometimes even within the same species [4,5]. The size of known angiosperm mitogenomes varies from 66 kbp in *Viscum scurruloideum* [6] to up to 11.3 Mbp in *Silene conica* [7]. Plant mitochondrial DNA (mtDNA) contains many repeats as well as inserts of nuclear and chloroplast origin, which makes mitogenome assembly difficult [8]. However, the development of long-read third-generation sequencing technologies (PacBio in particular) improves and simplifies assembly of such complex molecules, greatly facilitating the research of plant mitogenomes [9,10,11,12,13].

Due to the development of second- and third-generation sequencing technologies, the number of fully sequenced mitogenomes of a diverse spectrum of plants is growing fast, going beyond a set of economically important edible species [14]. Though to date, only 12 complete mitogenomes of species from the family Brassicaceae are presented in the GenBank genome database and only one of them from the genus Capsella—*C. rubella* (one of the *C. bursa-pastoris* progenitor species) [15]. In this work, we have assembled the complete mitogenome of *C. bursa-pastoris* using the single-molecule real-time (SMRT) PacBio sequencing technology. We have studied its gene profile and have analyzed its sequence and structure in comparison to currently available complete mitogenomes of the closely related species *C. rubella* and *A. thaliana* We have also investigated RNA editing sites using RNAseq data obtained from rRNA depleted total RNA of *C. bursa-pastoris*, and have identified several long (more than 300 bp) open reading frames (ORFs), the expression of which is supported by the RNAseq data.

## 2. Results

### 2.1. Sequencing and Assembly of the Complete Mitogenome of C. bursa-pastoris

*Capsella bursa-pastoris* total DNA was sequenced, and raw data had been prepared for assembly, resulting in 1,687,990 circular consensus sequencing (CCS) reads with an average read length of 7948 bp (61–26,706 bp) and an average quality of 83. Reads were aligned to the *C. bursa-pastoris* cpDNA reference sequence, and those that mapped with less than 5% divergence had been removed to avoid the interfering of cpDNA reads in the mitogenome assembly. Due to the length of CCS reads (~8 kbp or longer on average), mitogenome reads that contain chloroplast inserts have also passed the filter. After filtration, a 10% subsample has been isolated (145,590 reads with the average read length of 7964 bp) and used to assemble mitogenome, as downsampling often improves organelle assembly (e.g., [16,17]). One contig was identified as of a mitochondrial origin based on BLASTn search. It was circularized by merging ~9 kbp repeats at its ends. Then all CCS reads were mapped using minimap2 on the circularized contig to check the correctness of the assembly, resulting in 90,104 reads mapped to the contig covering 100% of its bases with an average coverage of 392. Thus, 287,799 bp long circular contig with an overall GC content of 44.74% has been identified as *C. bursa-pastoris* mtDNA. 

### 2.2. Repeats and Structural Variation Analysis

The sequence of the *C. bursa-pastoris* mitogenome contains 73 direct and 68 inverted repeats ranging from 28 to 854 bp in length with a minimum identity of 80% (Appendix A). In total, repeats occupy 8.2% of the mitogenome. To identify structural variants possibly emerging from these repeats, all CCS reads were mapped to the mitogenome using NGMLR, and then Sniffles was used to identify structural variants in the alignment, and was restricted to search only variants supported by at least 2 reads. Despite a large number of repeats found in the mitogenome, only 9 supported structural variants, corresponding to 8 repeats, have been found (Table 1).

The number of reads that supports the structural variant existence is proportional to the Bit score of the MEGABLAST algorithm. Generally, the longer the repeat and the higher the nucleotide similarity between its units, the more often recombination will occur. Thus, three repeats with the highest Bit score (Rep_1, Rep_2, Rep_3) give the three most frequent structural variants. For those structural options that are represented by a small number of reads, the correlation with the Bit score is weaker, because it increases the probability that the number of supporting reads of those structural variants is random. In total, 12,019 reads were mapped using NGMLR software to the *C. bursa-pastoris* mitogenome with the average coverage of 332. It means that even the most frequent of the structural variants could be found in only ~4% of the mitogenome copies. Though, despite the low representation, some of the structural variants lead to extensive changes in the mitogenome structure. For example, Rep_2 inverts the ~50 kbp long fragment of the mitogenome, and Rep_7 deletion splits it into two ~166 and ~120 kbp long molecules.

The full-genome alignment of the *C. bursa-pastoris* and *C. rubella* mtDNA sequences has shown that their mitogenomes are mostly collinear, except for two 47,173 bp and 5021 bp long inversions (Figure 1). The inversions are localized at sites 106,923–154,095 bp and 49,308–54,328 bp in the *C. bursa-pastoris* mitogenome and their borders lie within repeats Rep_4 and Rep_1, correspondingly.

CCS reads were mapped to *C. rubella* and two *C. bursa-pastoris* mtDNA sequences—one with an originally assembled sequence and the second where inversion regions have been manually replaced by reverse-complementary ones. Alignment of CCS reads to the artificial inversion region would reveal the presence of another *C. bursa-pastoris* mtDNA isoform collinear to the *C. rubella* mtDNA if it exists in the mitochondria. Mapping has shown that the uniformity and depth of coverage decrease at the borders of inversions of the artificially created isoform of the *C. bursa-pastoris* mtDNA (Appendix A). As for *C. rubella*, mapping has shown zero coverage at the borders of the inversions (Appendix A), which indicates that the originally assembled version of the *C. bursa-pastoris* mtDNA is its primary form. To investigate the inversions further, we have checked their presence in another progenitor species of *C. bursa-pastoris*—*C. orientalis*. Illumina paired-end reads from the GenBank WGS data of *C. orientalis* were mapped to the *C. bursa-pastoris* and *C. rubella* mtDNA sequences. Mapping results have shown that *C. orientalis* mitochondrial reads better align to the *C. bursa-pastoris* than to the *C. rubella* mtDNA across the entire sequence. *Capsella orientalis* reads aligned uniformly with high coverage at the borders of inversions to the *C. bursa-pastoris* mtDNA, while it failed to do so with the *C. rubella* mtDNA (Appendix A). Thus, we suggest that *C. orientalis* is the maternal progenitor of *C. bursa-pastoris*, as mitochondria are primarily maternally inherited.

### 2.3. Gene Content of the C. bursa-pastoris Mitogenome and Comparison with C. rubella and A. thaliana

The mitochondrial genome of *C. bursa-pastoris* contains 32 protein-coding genes, 3 rRNAs, 8 ORFs of unknown function, and 25 tRNAs corresponding to 15 amino acids (Figure 2). Comparison of the *C. bursa-pastoris* mitogenome gene content with *C. rubella* and *A. thaliana* (Appendix A) has shown that the set of protein-coding genes and rRNA in these plants is identical, except for two copies of the *atp6* gene in *A. thaliana*. However, some of the genes differ in both length and sequence between investigated species. *Arabidopsis thaliana* has two copies of the *atp6* gene, one of which is entirely different from the *atp6* genes of *C. rubella* and *C. bursa-pastoris*. The other copy differs from genes in *C. rubella* and *C. bursa-pastoris* by several non-synonymous substitutions and an indel spanning from 49 to 93 position in the gene nucleotide sequence. This indel makes the *atp6* gene of *A. thaliana* longer, and its nucleotide sequence encodes 15 amino acids instead of 5 amino acids in corresponding sequences of *C. bursa-pastoris* and *C. rubella*. *C. rubella* Current annotation of *C. rubella* states that the *atp9 gene* is longer than its orthologs in *C. bursa-pastoris* and *A. thaliana* by 33 bp and starts from ATG codon upstream of the ATG start codon common for this gene among Brassicaceae. The *ccmFC* gene of *C. rubella* is 30 bp longer than its orthologs in *C. bursa-pastoris* and *A. thaliana*. RNA editing creates a TGA stop codon that ends the *ccmFC* gene sequence in *A. thaliana* and *C. bursa-pastoris*. However, in *C. rubella* it continues further downstream to the TAA stop codon. There is no information on RNA editing in *C. rubella*, so the difference in length of the *ccmFC* genes could be due to incorrect annotation of this gene in the GenBank record of the *C. rubella* mitogenome. The *ccmFN1* gene of *A. thaliana* is 6 bp longer than its orthologs in *C. rubella* and *C. bursa-pastoris* due to an insert at the beginning of the gene. The *matR* gene of *C. rubella* is 60 bp longer at the 5′ end than its orthologs in *A. thaliana* and *C. bursa-pastoris*, and the annotation states that its start codon has not been determined. However, except for these additional nucleotides, gene sequences are similar for all Brassicaceae species. They have a common ATG start codon at the same position, including *C. rubella*, which allows us to suggest the incorrect annotation of this gene in the GenBank record of the *C. rubella* mitogenome. The *mttB* gene of *A. thaliana* is 27 bp longer than *mttB* in *C. rubella* and *A. thaliana* due to a frameshift caused by additional C nucleotides in the poly(C) sequence at the 3′ end of the gene, changing the TAG stop codon into CCT and moving the termination signal further downstream to the TGA stop codon. The start codon of the *mttB* gene is not determined, and both variants of annotation, as in *A. thaliana*, or further upstream, as in *C. rubella,* could be found in the Brassicaceae annotations of this gene. The *rpl2* gene of *C. rubella* is longer due to 3 bp and 21 bp long inserts in its sequence compared to the *rpl2* genes of *A. thaliana* and *C. bursa-pastoris*. 

Ribosomal RNAs 5S and 18S are identical in sequence between all three analyzed mitochondrial genomes, but sequence of the 26S rRNA of *A. thaliana* differs by three single nucleotide substitutions from its orthologs in *C. bursa-pastoris* and *C. rubella*. 

The set of tRNA is different between species, mostly due to the duplication of some tRNA in repeats (e.g., *C. bursa-pastoris* has duplication of the *trnY*-*GUA* + *trnS*-*GCU* region, *trnY*-*GUA*, and *trnK*-*UUU*). However, despite the differences in the number of tRNA between species, all of them represent the same set of 15 amino acids. According to the BLASTn search of the mitochondrial DNA against chloroplast DNA of *C. bursa-pastoris*, sequences of six of these tRNAs align with more than 95% identity. Additionally, they have been identified by MITOFY and CPGAVAS2 annotation software as the same tRNAs in both genomes, which allowed us to suggest that these tRNAs are of chloroplast origin: *trnS*-*GGA*, *trnM*-*CAU*, *trnW*-*CCA*, *trnD*-*GUC*, *trnH*-*GUG*, and *trnN*-*GUU*. Besides tRNA, inserts with chloroplast gene sequences and intergenic regions have also been found: *psbA*, *rpoB*, *psbD*-*psbC*, *psaB*-*psaA*, *rbcL*, and *ycf1*.

It is worth noting that rearrangements between *C. bursa-pastoris* and *C. rubella* mitogenomes that have been described earlier contain the intergenic region between two *trnY-GUA* tRNAs inside the ~5 kbp inversion and several genes inside the ~47 kbp inversion: *cox3*, the second and third exon of *nad1* and third exon of *nad5* trans-spliced genes, *ccmFN1*, *ccmC*, *atp4*, *nad4L*, two *trnS-GCU* tRNAs, three *trnY-GUA*, two *trnP-UGG*, tRNA *trnC-GCA*, *trnN-GUU*, and *trnE-UUC*.

In addition to known mitochondrial genes identified by MITOFY, 8 ORFs longer than 300 bp have also been found with expression supported by RNAseq data. According to the BLASTp results, most of these ORFs encode hypothetical proteins containing amino acid sequences similar to *cox2*, *atp9*, *atp6*, and *rpl2* genes of *A. thaliana* and other plants. One of the ORFs is a fragment of the chloroplast insert and corresponds to the beginning of the plastid *ycf1* gene diverged from it by 3 non-synonymous amino acid substitutions and a premature stop codon. All found ORFs have transmembrane regions and protein domains predicted by the InterProScan web-service, except *orf107* and *orf110*, which have no distinguishable protein features consistent with the results of the BLASTp search (Table 2).

The BLASTn search of the ORF nucleotide sequences against the *C. rubella* and *A. thaliana* mtDNA sequences has shown that only *orf197, orf107,* and *orf110* fully align with 100% identity to the *C. rubella* mtDNA. The rest of the ORFs align with more than 90% identity to both *C. rubella* and *A. thaliana* sequences, with better alignment scores for the *C. rubella* mtDNA. Though similar nucleotide sequences of most of the ORFs could be found in mitogenomes of related species, the INDELs and SNPs in *orf284* and orf290 changed their amino acid sequences significantly, resulting in a frameshift or premature stop codon. Thus, *orf284* and orf290 could be considered as ORFs specific to *C. bursa-pastoris*. 

### 2.4. RNA Editing

Ribosomal RNA depleted total RNA of *C. bursa-pastoris* was sequenced, generating 43,197,038 reads with an average length of 84 bp. Reads were mapped using the HISAT2 resulting in 28.8% of all reads mapped to the mitogenome sequence. Read alignment analysis has shown that there are 377 RNA editing sites (Appendix A), and only C to U transition events have been found. Only 21 of the discovered substitutions are synonymous, while the predominant part of editing events causes amino acid changes in encoded proteins. Most frequently, amino acids have been replaced by the leucine (164 cases), and less by phenylalanine (60 cases). In 15 cases, two editing sites have been identified in the same codon forming non-synonymous amino acid substitution. Editing sites have been found in all protein-coding mitochondrial transcripts, except *cox1* and *atp8*. Ribosomal proteins (except *rps4*) and ATPase subunits have a relatively small number of RNA editing derived substitutions (1–8 sites), while the transcripts of NADH dehydrogenase subunits (*nad1*, *nad2*, *nad4*, *nad5*, *nad7*) and cytochrome c biogenesis genes (*ccmB*, *ccmC*) have been significantly edited (19–30 sites; Figure 3).

## 3. Discussion

Size and GC content of the *C. bursa-pastoris* mitogenome are consistent with the characteristics of the Brassicaceae family mitogenomes (220–368 kbp and 44.74–45.33%, respectively) and almost identical to its closest relative *C. rubella* (287,799 bp vs. 287,405 bp and 44.74% equal for both). The presence of multiple repeats of various lengths is a characteristic feature of the plant mitochondrial DNA. With rare exceptions (e.g., *Silene conica* [7]), the mutation rate in the mitochondrial DNA of Embryophyta is much lower than in nuclear or chloroplast DNA, and diversity in the organization of plant mitochondrial genomes is provided by frequent recombination events at the repeat sites [18,19]. Due to high recombination frequency, plant mitochondrial genomes have a dynamic structure and are represented in mitochondria in various configurations (master ring, sub-rings, linear molecules) and stoichiometry. Most angiosperms have both short (up to 1 kbp) and long (from 1 kbp to more than 10 kbp) repeats in mitochondrial DNA, and Brassicales plants have at least one repeat longer than 1 kbp (except *Batis maritima*) [20]. Interestingly, no repeats longer than 1 kbp have been found in *C. bursa-pastoris*, as well as in *C. rubella*. Recombination across long repeats usually leads to the multipartite organization of the plant mitochondrial genome, where mtDNA isoforms exist in approximately equal stoichiometry. Whereas, recombination between short repeats occurs sporadically, forming low-copy isoforms of rearranged plant mitochondrial DNA (“sublimons”) [5,21]. Given the lack of long repeats and low CCS long-read support for the found structural variants, it could be assumed that the *C. bursa-pastoris* mitogenome is not in a multipartite form and a ~288 kbp long circular molecule is its main conformation. 

Two large inversions, the boundaries of which lie within 626 and 865 bp long inverted repeats, distinguish the mitochondrial genome of *C. bursa-pastoris* and *C. orientalis* from *C. rubella*. Due to the maternal inheritance of mitochondria, it could be assumed that *C. orientalis* is the maternal progenitor of the two progenitor species of *C. bursa-pastoris*. This is also indirectly confirmed by the higher coverage and better quality of the mapping of *C. orientalis* reads to the *C. bursa-pastoris* mitochondrial genome. This conclusion is consistent with the conclusions of the phylogenetic studies of the *C. bursa-pastoris* origin based on chloroplast DNA [2,22].

Plant mitochondrial genomes are redundantly large and consist of protein-coding genes, rRNA, and tRNA, interspersed with long intergenic regions, introns of cis- and trans-spliced genes, and many repeat sequences. A common ancestor set of genes of the angiosperms consisted of 41 genes: 24 core genes majorly involved in cellular respiration, 15 ribosomal genes, and succinate dehydrogenase complex subunits *sdh3* and *sdh4* [14,23]. According to the GenBank genome database, most of the flowering plants have preserved almost all the genes of this set in their mitogenomes. In the angiosperms, rRNA is usually represented by three genes—small 18S, large 26S, and 5S subunits, and tRNA varies significantly in number (from 0 to 44) and origin (mitochondrial or chloroplast). In the mitochondrial genome of *C. bursa-pastoris,* 32 protein-coding genes out of 41 have been detected, which means that the missing ribosomal and succinate dehydrogenase genes most likely have been transferred to the nuclear genome as it frequently happened during the evolution of angiosperms [24]. *Capsella bursa-pastoris* set of genes completely coincides with the set of *C. rubella* and *A. thaliana*, except for the second copy of the *atp6* gene in *A. thaliana*. It is known that plant mitochondrial DNA transfer bits of its sequence to the nuclear DNA (and rarely to the chloroplast DNA), and incorporates some of the nuclear and chloroplast DNA sequences in return. Angiosperms on average have 3–6% of plastid DNA in their mitogenome, of which only tRNAs are functional after transfer [14]. Among 25 tRNA of *C. bursa-pastoris,* six were found to be of chloroplast origin. Many seed plants, including *A. thaliana* and *Brassica napus*, have *trnS-GGA*, *trnM-CAU*, *trnW-CCA*, *trnD-GUC*, *trnH-GUG*, and *trnN-GUU* of chloroplast origin [25]. Including fragments of genes and intergenic spaces, the plastid inserts make 1.3% of the *C. bursa-pastoris* mitochondrial genome, which is close to 1% in *A. thaliana* [14]. Six out of eight ORFs, expression of which supported by RNAseq, in the *C. bursa-pastoris* mitochondrial genome contain transmembrane regions, and four of them are chimeric sequences of *cox2*, *atp6*, and *atp9* genes. ORFs containing chimeric sequences of these genes and transmembrane regions are known to cause cytoplasmic male sterility (CMS) in a wide array of plant species [26,27]. Thus, we suggest that found *C. bursa-pastoris* ORFs could be CMS-associated under the control of nuclear restorer-of-fertility (Rf) genes. Though, this conclusion needs further investigation.

It is notable, with few exceptions, that all editing sites that have been discovered in *C. bursa-pastoris* are the same as in *A. thaliana* mitogenome. Somewhat conservative status of RNA editing in mitochondrial genes has been previously described for Brassicaceae species *B. napus* and *A. thaliana* [28]. While comparing editing conversion of *ccmFN*, *cob*, *mttB*, *nad2*, and *nad4,* several SNPs in *C. bursa-pastoris* have been found in the RNA editing sites of *A. thaliana*. Thus, non-edited triplets in *C. bursa-pastoris* are encoding the same amino acids as the edited ones in *A. thaliana*. These data emphasize the evidence that RNA editing could be considered as the mechanism for restoring conserved codon identities that have been lost on the DNA level [29]. Among the essential editing events, those resulting in the start and stop codons formation should be highlighted. The start codon ATG in *nad1* is formed by RNA editing, which is common for Brassicaceae [15,30,31]. As for the stop codons, alteration in *ccmFC* transcript creates a stop codon TGA as in *A. thaliana*, though for some Brassicaceae, including *C. rubella*, transcript stop is suggested further downstream at the TAA stop codon [15,30,31]. Also, an internal stop codon formation (Gln8Ter) at the N terminus of the protein encoded by *rpl16* has been identified. This C to U conversion assumedly leads to the shortage of a translated region of *rpl16*. In *B. napus*, as well as in *A. thaliana*, RNA editing creates a stop codon at the 21st amino acid from the start, leading to the suggestion of an alternate site of transcription initiation of *rpl16* or its pseudogenization [28].

## 4. Materials and Methods

### 4.1. DNA Extraction and Sequencing

Freshly harvested leaves of *C. bursa-pastoris* (the line ‘msu-wt’ [32]) were quick-frozen in liquid nitrogen and transferred on dry ice to the DNA Link laboratory (South Korea, Seoul) where DNA was extracted from the sample and sequenced using PacBio RS II. High-precision CCS reads were prepared from raw sequencing data by the DNA Link as well as by using the Circular Consensus Sequencing (CCS) application from the SMRTLink v8.0 software with default parameters.

### 4.2. Mitochondrial and Chloroplast Genome Assembly

Chloroplast reads were filtered out by mapping CCS reads to the *C. bursa-pastoris* chloroplast genome reference from the NCBI GenBank database (RefSeq NC_009270.1) using Minimap2 2.17-r954-dirty [33] with a set of predefined parameters for mapping PacBio CCS reads (“-ax asm5”) and keeping only unmapped reads. Then a 10% subsample of the filtered reads was used for the assembly, using Canu v1.8 [34] without read correction (“-assemble -pacbio-corrected correctedErrorRate = 0.005 minOverlapLength = 500 minReadLength = 1000”). Candidate mitochondrial contigs were identified by sequence identity to *C. rubella* mtDNA (Genbank accession: MH624151.1) using BLASTn 2.9.0 + [35] alignment. The contig with the highest identity and coverage was selected as the primary candidate mtDNA for further analysis.

Chloroplast reads that were filtered out previously were subsampled to 20,000 reads and assembled with Canu v1.9 [34] without read correction. The chloroplast contig was identified using BLASTn search against *C. bursa-pastoris* chloroplast genome reference (RefSeq accession: NC_009270.1). It contained a full long single copy (LSC), short single copy (SSC), and first inverted repeat (IRa) regions and border fragments of the second inverted repeat (IRb) at the ends of the contig. The chloroplast genome was completed and circularized by filling the unassembled IRb fragment between the contig’s ends with the corresponding IRa sequence. The assembled chloroplast genome is longer than GenBank reference (RefSeq NC_009270.1) by 63 bp and differs by 84 SNPs and 33 INDELs.

### 4.3. Repeats and Structural Variation Analysis

The *C. bursa-pastoris* mtDNA was aligned against itself using the online version of MEGABLAST [35], with the e-value threshold set to 10^−5^ to locate repeats. Analysis of structural variants was carried out as described elsewhere [36] by mapping the CCS reads using NGMLR 0.2.7 with the default settings to the mtDNA, and finding the structural variants in the alignment with Sniffles 1.0.11. Structural variants had to be supported by at least 2 reads. The *C. bursa-pastoris* mtDNA was also aligned against assembled cpDNA (assigned GenBank accession MT040199) using BLASTn 2.9.0 + [35] with the default settings to locate inserts of chloroplast origin in the mitochondrial genome.

The sequence of *C. bursa-pastoris* mtDNA was compared with the *C. rubella* mtDNA (RefSeq: NC_042883.1) using Mauve snapshot_2015-02-13 build 0 [37] genome-wide alignment. Additionally, reads from the Russian population of *C. orientalis* (SRA SRR8904471) were mapped to both *C. rubella* and *C. bursa-pastoris* mtDNA to check the presence of identified structural variants in both progenitor species of *C. bursa-pastoris* by mapping quality assessment. Visual analysis of the alignments was performed using IGV 2.7.2 [38].

### 4.4. RNA Extraction, Sequencing, and Analysis

Total RNA was extracted from the fresh leaf material of *C. bursa-pastoris* (the line ‘msu-wt’ [32]) using the RNeasy Mini Kit (Qiagen, Hilden, Germany). RNAseq libraries were prepared with Zymo-Seq RiboFree Total RNA Library Kit (Zymo Research, Orange, CA, USA), and single-read sequencing was performed on Illumina NextSeq 500 using the NextSeq 500/550 High Output Kit v2 (75 cycles) (Illumina, Mountain View, CA, USA).

RNAseq analysis was conducted by mapping reads to the *C. bursa-pastoris* annotated mitogenome using HISAT2 2.1.0 [39] and the visual analysis of alignment with the IGV 2.7.2 [38]. RNA editing sites were identified using variant calling results of the mapped data generated by bcftools 1.9 (with settings “mpilepup -I -B -d 8000” and then “call -m -V indels -Ov”) [40,41] and the REDO script [42] with settings “-d 30 -c 10 -s 0 -a 0”. To correctly identify the percent of reads supporting each editing site, all RNAseq reads were mapped to the original mtDNA sequence and modified the mtDNA with all predicted substitutions present in the sequence using CLC Genomics Workbench 9.5.4 (CLC bio, Aarhus, Denmark) with parameters allowing only unique mapping, no mismatches, and 100% of the read length mapped. To calculate read depth at each RNA editing position in both references, samtools 1.9 [43] with “depth” command was used. All editing sites were also checked manually using IGV 2.7.2 [38] and compared with *A. thaliana* editing sites.

### 4.5. Genome Annotation

*Capsella bursa-pastoris* mitogenome was annotated using a MITOFY web server [44], and the chloroplast genome was annotated using a CPGAVAS2 web server [45], with subsequent manual verification and, if necessary, correction of the found gene boundaries by comparison of nucleotide and amino acid sequences with corresponding ortholog sequences in *A. thaliana* and *C. rubella*. The annotated mitochondrial and chloroplast genomes of *C. bursa-pastoris* were submitted to the NCBI GenBank database (assigned accessions MN746809 and MT040199, respectively). ORFfinder NCBI web-service was used to locate ORFs in the mitogenome with a length no less than 300 bp in regions where ORFs expression was supported by RNAseq data (read coverage ≥ 1000 within ORF, while the mitogenome median coverage is 73). The mitochondrial genome map was created with Circos v. 0.69–9 [46].

## 5. Conclusions

A single circular master molecule represents the mitogenome of *C. bursa-pastoris*. It is 287,799 bp long, contains 32 protein-coding genes, 3 rRNAs, and 25 tRNAs, which coincides with the average length and gene profile of the other known complete mitogenomes of the Brassicaceae species. Investigation of the RNA editing sites in coding regions of the mitogenome using RNAseq data has revealed 377 C to U transitions, most of which are the same editing sites as in *A. thaliana*. Nearly identical edited amino acid sequences of the genes, with both matched and mismatched RNA editing sites in *C. bursa-pastoris* and *A. thaliana*, provide additional evidence of the RNA editing conserved nature in the Brassicaceae species. We have also identified eight new long (more than 300 bp) ORFs with their expression confirmed by RNAseq data (orf107, orf110, orf161, orf197, orf230, orf263, orf284, orf290). Most of them contain transmembrane regions and chimeric sequences of mitochondrial and plastid genes, which resemble characteristics of the CMS-associated genes, though this connection requires further investigation. Analysis of the structural variants of the *C. bursa-pastoris* mitogenome has revealed that a single circular master molecule is its primary form, while other possible structural variants are almost absent in the mitochondria. We also suggest that *C. orientalis* is a maternal progenitor species of *C. bursa-pastoris* based on the presence of two large inversions in the mitogenome of *C. bursa-pastoris* when compared to *C. rubella*, and lack of thereof when compared to *C. orientalis*. Data obtained in the current research could be useful for future investigations associated with the organization of plant mitochondrial DNA and phylogenetic studies of angiosperms and the family Brassicaceae in particular.

## Figures and Tables

**Figure 1 plants-09-00469-f001:**
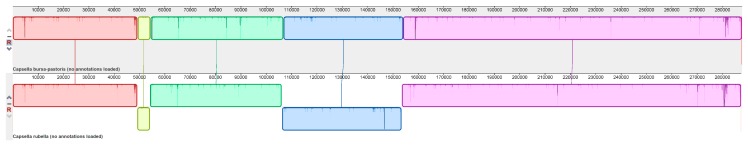
Full-genome alignment of mtDNA of *C. bursa-pastoris* (top) and *C. rubella* (bottom). Inversion block 47,173 bp is blue, and inversion block 5021 bp is yellow.

**Figure 2 plants-09-00469-f002:**
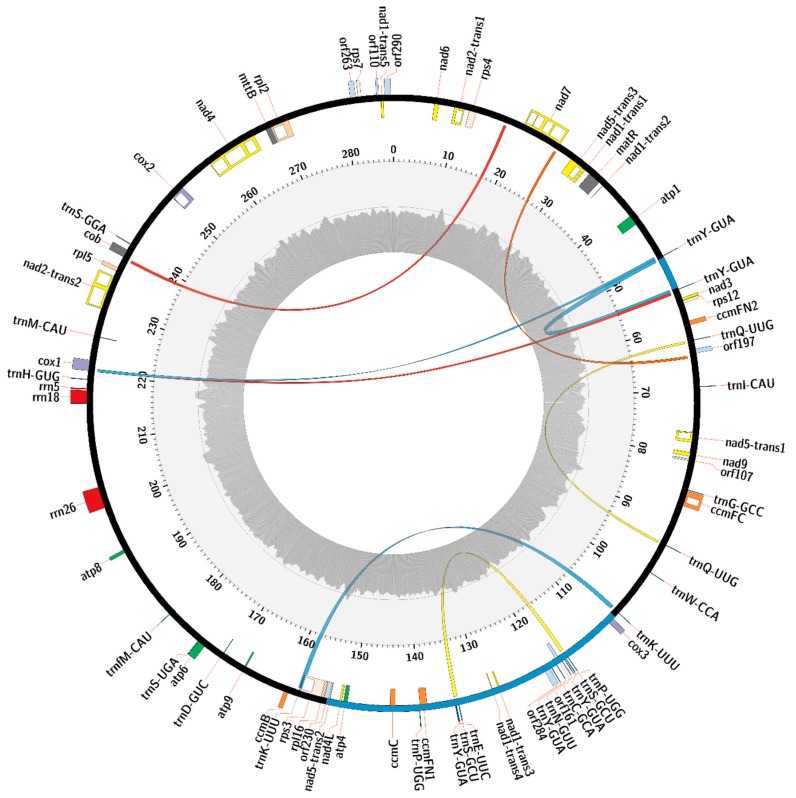
Map of the mitogenome of *C. bursa-pastoris*. Colored blocks on circular axis denote genes, and white blocks within them are introns. The gray histogram on the inner ring shows the guanine-cytosine (GC) composition with the thin dark gray line denoting 50% GC content. Arcs indicate repeats associated with structural variants: deletions are red, duplications are yellow, inversions are blue, and the orange arc is a repeat, which corresponds to both deletion and duplication. Two blue regions on the circular axis represent inversions in comparison with the *C. rubella* mitogenome.

**Figure 3 plants-09-00469-f003:**
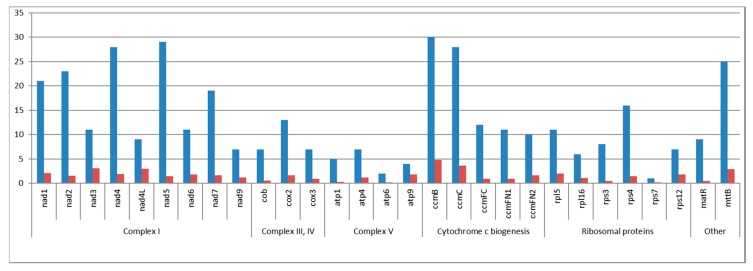
The absolute number of RNA editing substitutions per gene (blue bars) and the relative number of RNA editing substitutions by gene length normalized to 100 bp (red bars).

**Table 1 plants-09-00469-t001:** The structural variants in the mitogenome of *C. bursa-pastoris* supported by at least two circular consensus sequencing (CCS) reads.

Repeat ID	Structural Variant Type	Number of Supporting CCS Reads	Repeat Length (bp)	First Repeat Unit Position (bp)	Second Repeat Unit Position (bp)	Repeat Units’ Sequence Identity
Rep_1	inversion	13	854	54,219–55,072	49,506–48,653	98.9%
Rep_2	inversion	8	635	158,147–158,781	106,515–105,881	100.0%
Rep_3	duplication	6	538	134,019–134,556	116,143–116,679	99.8%
Rep_7	deletion	5	418	220,620–221,037	54,655–55,072	99.8%
Rep_10	inversion	4	420	220,623–221,042	49,067–48,648	97.9%
Rep_11	duplication	3	356	64,729–65,084	26,047–26,402	100.0%
Rep_12	duplication	3	327	93,763–94,089	62,446–62,772	100.0%
Rep_9	deletion	2	404	238,171–238,573	17,371–17,774	99.8%
Rep_11	deletion	2	356	64,729–65,084	26,047–26,402	100.0%

**Table 2 plants-09-00469-t002:** RNAseq supported open reading frames (ORFs) of the *C. bursa-pastoris* mitogenome.

Name *	InterProScan Predictions	BLASTp Similarity
*orf290*	Cytochrome c oxidase, subunit II (*cox2*) domain with 2 transmembrane regions within	*cox2*
*orf197*	1 transmembrane region	*atp6*
*orf107*	Nothing found	hypothetical protein
*orf161*	Member of Protein TIC214 (*ycf1*) InterPro family. Ycf1 domain and 4 transmembrane regions within the domain	*ycf1*
*orf284*	Signal peptide (located 1–17 aa) and 2 transmembrane regions	*atp9*
*orf230*	Member of Ribosomal protein L2 (*rpl2*) InterPro family. Ribosomal_L2 domain and 1 transmembrane region	*rpl2*
*orf263*	Member of ATP synthase, F0 complex, subunit C (*atp9*) InterPro family. ATP synthase, subunit C, isoform a domain and 3 transmembrane regions within the domain	*atp9*
*orf110*	Nothing found	hypothetical protein

*—ORFs are named by their amino acid length.

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
