# Peer review of "Assembly and Analysis of the Complete Mitochondrial Genome of *Capsella bursa-pastoris"

_plants, 2020, doi:10.3390/plants9040469_

Round 1

Reviewer 1 Report

Omelchenko et al. report assembly and analysis of C. bursa-pastoris mitogenome.

C. bursa-pastoris is an annual weed belonging to Brassicaceae family which relevance in angiosperms’ allopolyploidization studies is highlighted. Plant mtDNA is a very complex and dynamic molecule; it is a valuable source of genetic information therefore the present paper is a useful addition to the literature on plant mitochondrial genomes.

The paper is well-written and organised. However I have few suggestions.

  • C. bursa-pastoris mitogenome is not yet available in GenBank.
  • C. bursa-pastoris mitochondrial DNA was compared to Capsella rubella and A. thaliana mtDNA sequences available online. Authors found that mitochondrial DNA of C. bursa-pastoris and C. rubella were collinear except for two inversions of 5 and 47 Kbp, respectively. They verified the inversion by manually reverse and complement the inversions and comparing the mtDNA from the two species. Zero coverage resulted at borders of inversions, however experimental evidence is missing. A PCR analysis should be performed.
  • Line 114 states that the set of protein-coding genes in C. bursa-pastoris, C. rubella and A. thaliana is identical, however some slight differences in gene length is observed and should be mentioned (e.g. mttB).
  • Line 124 states that six tRNAs are of plastid origin. It would be interesting to verify whether these are effectively expressed or are pseudogenes.
  • Line 141 Why don’t you refer to these proteins as chimeric proteins?
  • The authors also provide info on RNA processing describing RNA editing events in coding regions. Also, I would suggest extending this analysis also to non-coding regions such as tRNAs and introns. The latters are important targets of the editing machinery (e.g. in all flowering plants introns are members of the group II ribozyme family).
  • Line 219 Could you please explain why you consider only four out of six orfs containing transmembrane domains as chimeric proteins?
  • Line 227 A “.” is missing at the end of the sentence.
  • Table 2: as orfs are named by their length, I would delete the “length, AA” column.
  • The authors should provide an outline of C. bursa-pastoris mitogenome utilization in future studies in “Conclusions” section

Author Response

Response to Reviewer 1 Comments

We would like to thank the reviewer for the thorough and helpful review. Line numbers in responses correspond to lines in the revised version of the manuscript.

Point 1: C. bursa-pastoris mitogenome is not yet available in GenBank.

Response 1: We`ve submitted C. bursa-pastoris mitogenome to GenBank with the release date. It will become available under accession MN746809 as soon as the article with the accession is published or on the release date, whichever happens first.

Point 2: C. bursa-pastoris mitochondrial DNA was compared to Capsella rubella and A. thaliana mtDNA sequences available online. Authors found that mitochondrial DNA of C. bursa-pastoris and C. rubella were collinear except for two inversions of 5 and 47 kbp, respectively. They verified the inversion by manually reverse and complement the inversions and comparing the mtDNA from the two sites. Zero coverage resulted at borders of inversions, however experimental evidence is missing. A PCR analysis should be performed.

Response 2: According to SMRT technology specs, the accuracy of CCS reads averages 99.8%, which is close to the accuracy of short-read sequencing. Given enough depth, high-throughput sequencing is better than PCR in the detection of patterns of the plant mitochondrial recombination (e.g., Alverson, A. J., Zhuo, S., Rice, D. W., Sloan, D. B., & Palmer, J. D. (2011). The mitochondrial genome of the legume Vigna radiata and the analysis of recombination across short mitochondrial repeats. PloS one, 6(1), e16404. https://doi.org/10.1371/journal.pone.0016404). Thus, we suggest that PCR confirmation, in this case, is not necessary, and CCS read mapping results are enough to support our conclusions.

Point 3: Line 114 states that the set of protein-coding genes in C. bursa-pastoris, C. rubella and A. thaliana is identical, however some slight differences in gene length is observed and should be mentioned (e.g. mttB).

Response 3: Yes, except for the second copy of the atp6 gene, C. bursa-pastoris, C. rubella, and A. thaliana share the same set of protein-coding genes, though some of them differ in sequence and length. We`ve added details about gene differences in the results (lines 137-163). We`ve also corrected miscalculated length of mttB and nad1 genes of C. bursa-pastoris in TableS2.

Point 4: Line 124 states that six tRNAs are of plastid origin. It would be interesting to verify whether these are effectively expressed or are pseudogenes.

Response 4: To correctly measure tRNAs expression and detect their modifications, specific tRNAseq library preparation protocols are required (e.g., see Wilusz, J. Removing roadblocks to deep sequencing of modified RNAs. Nat Methods 12, 821–822 (2015). https://doi.org/10.1038/nmeth.3516; Warren, Jessica M., Thalia Salinas-Giegé, Guillaume Hummel, Nicole L. Coots, Joshua M. Svendsen, Kristen C. Brown, Laurence Maréchal-Drouard, и Daniel B. Sloan. 2019. «Combining tRNA sequencing methods to characterize plant tRNA expression and post-transcriptional modification». BioRxiv, 790451.). Our RNAseq data is suitable only for RNA editing analysis and detection of expression of long ORFs and protein-coding genes. Though we have found that read coverage is above background level (>100) for several tRNAs (trnY-GUA, trnS-GCU, trnN-GUU, trnM-CAU, trnC-GCA), with the highest coverage (>1000) for trnN-GUU (this is one of the chloroplast origin), these results are unreliable, and we can`t include them in the manuscript.

Point 5: Line 141 Why don’t you refer to these proteins as chimeric proteins?

Response 5: We`ve studied these ORFs sequences in situ and detected their expression using RNAseq, but we haven`t observed them on the protein level. Thus, we can`t refer to them as proteins until we verify their translation experimentally.

Point 6: The authors also provide info on RNA processing describing RNA editing events in coding regions. Also, I would suggest extending this analysis also to non-coding regions such as tRNAs and introns. The latters are important targets of the editing machinery (e.g. in all flowering plants introns are members of the group II ribozyme family).

Response 6: Most of the tRNA and intron regions in our RNAseq data have insufficient coverage, close to the background level, so it is not possible to correctly perform RNA editing analysis and determine editing sites unambiguously.

Point 7: Line 219 Could you please explain why you consider only four out of six orfs containing transmembrane domains as chimeric proteins?

Response 7: All six ORFs contain transmembrane regions, but only four of them have protein domains, according to InterProScan results. We have not performed experiments to verify their translation, so we refrain from calling them proteins.

Point 8: Line 227 A “.” is missing at the end of the sentence.

Response 8: We`ve corrected this error.

Point 9: Table 2: as orfs are named by their length, I would delete the “length, AA” column.

Response 9: We`ve deleted the column and added a footer to the table that ORFs are named by their AA length.

Point 10: The authors should provide an outline of C. bursa-pastoris mitogenome utilization in future studies in the “Conclusions” section.

Response 10: We`ve added the “Conclusions” section after the “Materials and Methods” (lines 366-385).

Reviewer 2 Report

Here Omelchenko et al. use PacBio 3rd gen sequencing to sequence and assemble the mitogenome for a cosmopolitan annual weed. They combine this with some comparative analyses and RNAseq data to descriptively evaluate the mitogenome for gene content, repeats, structural variants, and RNA editing sites. Overall, the methodology seems sound and this mitogenome is surely a worthwhile addition to GenBank. Additionally, this study demonstrates another utility for PacBio long read data. However, I feel that the organization and presentation could be honed, so that justification of included methods is fully clear. I imagine this is partly a result of writing the manuscript in Intro-methods-results-discussion order, and then editing it to fit the journal’s results-first organization. As-is, I feel the reader would benefit greatly from more explicit summary of what will be done in this paper in the last paragraph of the introduction.

Additionally, while the manuscript reads fairly well, oddly structured sentences and poor tense choice distract from both the flow and the content. I have made comments about these issues in the first page or so, but they are present throughout the manuscript; editing by a native English speaker is required before this will be fit for publication.

SPECIFIC COMMENTS

Line 13. I would specify “plant mitochondrial genomes” here, since the same would not necessarily be true about animal mitogenomes.

Line 14. Should it be “C. rubella’s”?

Lines 36-37. Should read “Mitochondrial genomes (mitogenomes) are”.

Lines 38-39. I would suggest rephrasing as “…are highly variable molecules in both size and structure when compared with more conservative chloroplast genomes”. Also, I am not sure what is meant by the word “conservative”. Perhaps “conserved”, but that still isn’t entirely clear (i.e. conserved relative to what?).

Line 38. It might be beneficial here to add something along the lines of “(hereafter we explicitly refer to plant mitogenomes)”, as some statements (as brought up for line 13) would be incorrect for all animal mitogenomes.

Lines 40-41. I would suggest rephrasing: “The size of known angiosperm mitogenomes varies from 66 kbp…”

Lines 42-43. Rephrase: “Mitochondrial DNA (mtDNA) contains many repeats as well as inserts of nuclear and chloroplast origin, which makes mitogenome assembly difficult”

Line 53. The tense of this sentence is odd. Why not say “In this work, we assembled the complete mitogenome of C. bursa-pastoris using…”

Lines 54-55. Again, I find the tense of this sentence to be distracting.

Line 61. At line 43, the authors state that mitogenomes can have chloroplast inserts—It would be helpful to clarify how would these be identified using the stated methodology. Or be more explicit about “reads fully matching cpDNA were removed” or something along those lines.

Lines 61-62. I think a citation justifying the subsampling is warranted here.

Line 73. I find this sentence confusing.

Line 91. Is it possible to visualize the results of Table 1 on this figure? This would help to contextualize the last paragraph.

Line 101. “It was interesting” is too colloquial for scientific writing in my opinion.

Line 108. You “could” suggest? Either suggest or don’t suggest. There isn’t a middle ground there.

Figure S2. I don’t understand what is being shown in this figure and can hardly make out the tiny text in the screen captures. Why not quantify the mapping success instead of just showing screen captures? Or at least annotate these images so it is clear what is being demonstrated.

Line 140. Given that the results are presented first in this journal, more background needs to be provided about this RNAseq data—where did it come from? What species?

Line 156. Again, this data kind of comes out of nowhere. In my opinion, it would be better to introduce these additional datasets in the last paragraph of the intro. Given the title and abstract it’s clear that mitogenome sequencing is going to occur in this paper (from a reader’s perspective), but these additional aspects aren’t introduced at all.

Line 242. The general terminology is “flash frozen”.

Lines 244-245. How did DNA Link call consensus sequences for these CCS reads? This is a key part of data processing for this type of sequence data and needs to be specified.

Lines 256-263. While an interesting comparison, this chloroplast genome wasn’t mentioned in the results at all. Why include it if the results aren’t presented or discussed?

Line 270. Which cpDNA was used here? The assembly from PacBio data or the existing cpgenome?

Author Response

Response to Reviewer 2 Comments

We would like to thank the reviewer for the thorough review and constructive suggestions for improving the quality of the manuscript. Line numbers in responses correspond to lines in the revised version of the manuscript.

Point 1: As-is, I feel the reader would benefit greatly from more explicit summary of what will be done in this paper in the last paragraph of the introduction.

Response 1: We`ve added more details in the last paragraph of the introduction. lines 60-62: ”We have also investigated RNA editing sites using RNAseq data obtained from rRNA depleted total RNA of C. bursa-pastoris, and identified several long ORFs, expression of which is supported by the RNAseq data.”.   

Point 2: Additionally, while the manuscript reads fairly well, oddly structured sentences and poor tense choice distract from both the flow and the content. I have made comments about these issues in the first page or so, but they are present throughout the manuscript; editing by a native English speaker is required before this will be fit for publication.

Response 2: We`ve done our best to improve English style of the manuscript.

Point 3: Line 13. I would specify “plant mitochondrial genomes” here, since the same would not necessarily be true about animal mitogenomes.

Response 3: We`ve corrected sentence at lines 12-13 to “plant mitochondrial genomes”.

Point 4: Line 14. Should it be “C. rubella’s”?

Response 4: It is not common in scientific writing to use a possessive apostrophe with Latin scientific names. For example, it is almost always “A. thaliana genome” in articles, but not “A. thaliana`s genome”. We suggest it would be better to rephrase it to "the complete mitogenome of C. rubella"

Point 5: Lines 36-37. Should read “Mitochondrial genomes (mitogenomes) are”.

Response 5: We`ve corrected the sentence at lines 36-37 to “Mitochondrial genomes (mitogenomes) are”.

Point 6: Lines 38-39. I would suggest rephrasing as “…are highly variable molecules in both size and structure when compared with more conservative chloroplast genomes”. Also, I am not sure what is meant by the word “conservative”. Perhaps “conserved”, but that still isn’t entirely clear (i.e. conserved relative to what?).

Response 6: We have rephrased the sentence at lines 38-40 to “Plant mitogenomes are highly variable molecules in both size and structure in contrast to chloroplast genomes that have highly conserved quadripartite structure among land plants.”.

Point 7: Line 38. It might be beneficial here to add something along the lines of “(hereafter we explicitly refer to plant mitogenomes)”, as some statements (as brought up for line 13) would be incorrect for all animal mitogenomes.

Response 7: We`ve added “plant” in all sentences that refer to plant mitochondrial genomes throughout the manuscript.

Point 8: Lines 40-41. I would suggest rephrasing: “The size of known angiosperm mitogenomes varies from 66 kbp…”

Response 8: We have changed the sentence at lines 41-42 to “The size of known angiosperm mitogenomes varies from 66 kbp in Viscum scurruloideum [6] up to 11.3 Mbp in Silene conica [7]”.

Point 9: Lines 42-43. Rephrase: “Mitochondrial DNA (mtDNA) contains many repeats as well as inserts of nuclear and chloroplast origin, which makes mitogenome assembly difficult”

Response 9: We have rephrased the sentence at lines 43-44 to “Plant mitochondrial DNA (mtDNA) contains many repeats as well as inserts of nuclear and chloroplast origin, which makes mitogenome assembly difficult.”.

Point 10 & 11: Line 53. The tense of this sentence is odd. Why not say “In this work, we assembled the complete mitogenome of C. bursa-pastoris using…”. Lines 54-55. Again, I find the tense of this sentence to be distracting.

Response 10 & 11: We have rephrased sentences at lines 54-58 to “In this work, we have assembled the complete mitogenome of C. bursa-pastoris using the single-molecule real-time (SMRT) PacBio sequencing technology. We have studied its gene profile, analyzed its sequence and structure, in comparison to currently available complete mitogenomes of the closely related species C. rubella and A. thaliana.”.

Point 12: Line 61. At line 43, the authors state that mitogenomes can have chloroplast inserts—It would be helpful to clarify how would these be identified using the stated methodology. Or be more explicit about “reads fully matching cpDNA were removed” or something along those lines.

Response 12: We have added details and rephrased the sentence at line 69-75 to “Reads were aligned to the C. bursa-pastoris cpDNA reference sequence, and those that mapped with less than 5% divergence had been removed to avoid interfering of cpDNA reads in the mitogenome assembly. Due to the length of CCS reads (~8 kbp or longer on average), mitogenome reads that contain chloroplast inserts have also passed the filter. After filtration, a 10% subsample has been isolated (145,590 reads with the average read length of 7,964 bp) and used to assemble mitogenome, as downsampling often improves organelle assembly (e.g., [16,17]).”. Chloroplast inserts have been identified by BLASTn search of the mitogenome sequence against assembled plastid genome sequence of C. bursa-pastoris as it has been described in the results at lines 170-177 and in the “Materials and Methods” section. 

Point 13: Lines 61-62. I think a citation justifying the subsampling is warranted here.

Response 13: Downsampling is a common practice to save computational time and resources, and often improves organelle assembly due to significantly higher coverage of organelle reads (both mitochondrial and plastid) in comparison to nuclear reads in WGS data. We`ve added citations as an example in the sentence at lines 73-75.

Point 14: Line 73. I find this sentence confusing.

Response 14: We have split and rephrased the sentence at lines 85-91 to make it more clear: “To identify structural variants possibly emerging from these repeats, all CCS reads were mapped to the mitogenome using NGMLR, and then Sniffles was used to identify structural variants in the alignment, restricted to search only variants supported by at least 2 reads. Despite a large number of repeats found in the mitogenome, only 9 supported structural variants, corresponding to 8 repeats, have been found.”.

Point 15: Line 91. Is it possible to visualize the results of Table 1 on this figure? This would help to contextualize the last paragraph.

Response 15: Both Table 1 results and inversions from Figure 1 are visualized in Figure 2 on the circular map of the C. bursa-pastoris mitogenome. Repeats with structural variants supported by CCS reads from Table 1 are arcs of different colors, where color corresponds to the type of structural variant, and inversions from figure 2 are blue colored regions on the circular axis of the map. It is written in the caption of Figure 2, lines 182-185.

Point 16: Line 101. “It was interesting” is too colloquial for scientific writing in my opinion.

Response 16: We`ve rephrased the sentence at lines 122-124 to “To investigate inversions further, we have checked their presence in another progenitor species of C. bursa-pastorisC. orientalis.”.

Point 17: Line 108. You “could” suggest? Either suggest or don’t suggest. There isn’t a middle ground there.

Response 17: We`ve deleted “could” in the sentence at line 129.

Point 18: Figure S2. I don’t understand what is being shown in this figure and can hardly make out the tiny text in the screen captures. Why not quantify the mapping success instead of just showing screen captures? Or at least annotate these images so it is clear what is being demonstrated.

Response 18: Supplementary figures S1 and S2 show differences in the coverage of the read alignment to C. bursa-pastoris and C. rubella mtDNA in the regions of inversions. Unfortunately, the IGV browser software has limited settings for text format and image saving. However, we`ve increased font size for track names, enlarged coverage histogram, and added more information in the figure captions for Figure S1 and Figure S2.   

Point 19: Line 140. Given that the results are presented first in this journal, more background needs to be provided about this RNAseq data—where did it come from? What species?

Response 20: RNAseq data has been obtained from the sequencing of total RNA of C. bursa-pastoris depleted from ribosomal RNA. Detailed information is presented in the “Materials and methods” section, “4.4. RNA extraction, sequencing, and analysis” subsection.

Point 20: Line 156. Again, this data kind of comes out of nowhere. In my opinion, it would be better to introduce these additional datasets in the last paragraph of the intro. Given the title and abstract it’s clear that mitogenome sequencing is going to occur in this paper (from a reader’s perspective), but these additional aspects aren’t introduced at all.

Response 20: We have added more details on C. bursa-pastoris in the last paragraph of the introduction (see Response 1).

Point 21: Line 242. The general terminology is “flash frozen”.

 We`ve seen both “flash-frozen” and “quick-frozen” equally frequently (sometimes even “snap-frozen”) in articles referring to the liquid nitrogen freezing process. We think correction is not necessary as these terms are synonyms.

Point 22: Lines 244-245. How did DNA Link call consensus sequences for these CCS reads? This is a key part of data processing for this type of sequence data and needs to be specified.

Response 22: We have asked DNA Link for additional information on CCS data preparation and added their answer to the “Materials and Methods” section, “4.1. DNA extraction and sequencing” subsection, lines 304-305: “High-precision CCS reads were prepared from raw sequencing data by the DNA Link as well using Circular Consensus Sequencing (CCS) application from the SMRTLink v8.0 software with default parameters.”.

Point 23: Lines 256-263. While an interesting comparison, this chloroplast genome wasn’t mentioned in the results at all. Why include it if the results aren’t presented or discussed?

Response 23: Article is focused on the mitochondrial genome analysis, and the assembled chloroplast genome has been used mainly in our other studies that are not included in this work. Here it has been used only to locate chloroplast inserts in the mitogenome. We`ve wanted to use reference cpDNA from the same specimen as assembled mitogenome rather than a reference from the GenBank. The comparison in the “Materials and Methods” section is presented only to show a similarity between assembled and GenBank cpDNA sequences of C. bursa-pastoris.   

Point 24: Line 270. Which cpDNA was used here? The assembly from PacBio data or the existing cpgenome?

Response 24: We used our assembled cpDNA. We`ve added this information in the “Materials and Methods” section in the sentence at line 330.

Round 2

Reviewer 1 Report

The manuscript was significantly improved. The points previously highlighted were resolved and, when it was not possible, authors provided literature-based explanations. 

Reviewer 2 Report

The authors have addressed all of my original comments, and I think the manuscript reads better now (particularly with the few additions to the end of the intro to describe all of the contents). Just a handful of grammatical suggestions:

Line 68. “Was” instead of “has been”

Line 126. Spell out the genus name if starting a sentence with it. Line 252, 340 as well.

Line 129. “This indel makes…”

Line 136. “until the” instead of “till”

Line 139. “an insert”

Line 255. “on average”, and remove the “of” at the end of line.

Line 257. Spell out 6.

Line 299. “The contig…”

Line 302. “The chloroplast contig…”

Line 307. “The assembled…”

Line 340. “and the chloroplast genome”

Line 344. “genomes”